# ENHANCING ADVERSARIAL TRANSFERABILITY VIA COMPONENT-WISE TRANSFORMATION

## ABSTRACT

Deep Neural Networks (DNNs) are highly vulnerable to adversarial examples, which pose significant challenges in security-sensitive applications. Among various adversarial attack strategies, input transformation-based attacks have demonstrated remarkable effectiveness in enhancing adversarial transferability. However, current methods struggle with cross-architecture transferability, even when performing well within the same architecture. This limitation arises because, while models of the same architecture may focus on different regions of the object, the variation is even more pronounced across different architectures. Unfortunately, current approaches fail to effectively guide models to attend to these diverse regions. To address this issue, this paper proposes a novel input transformation-based attack method, termed Component-Wise Transformation (CWT). CWT applies interpolation and selective rotation to individual image blocks, ensuring that each transformed image highlights different target regions. Extensive experiments on the standard ImageNet and COCO datasets demonstrate that CWT consistently outperforms state-of-the-art methods across both CNN- and Transformer-based models.

## 1 INTRODUCTION

With the rapid development of deep learning, artificial intelligence has achieved significant progress in computer vision, finding widespread applications in tasks such as image classification (He et al., 2016), object detection (Ren et al., 2016), and semantic segmentation (Ronneberger et al., 2015). However, the adversarial vulnerability of deep learning models has gradually surfaced as a critical issue, limiting their deployment in real-world scenarios. Research shows that carefully crafted adversarial examples—created by adding imperceptible perturbations to normal input data—can cause deep learning models to make erroneous predictions (Szegedy, 2013; Goodfellow et al., 2014). This phenomenon not only affects model performance on standard test datasets but poses severe risks in high-stakes applications (Eykholt et al., 2018) such as autonomous driving and medical diagnostics.

In black-box scenarios, attackers lack access to the details of the target models and can only rely on a limited number of surrogate models to craft adversarial examples. Therefore, a critical challenge is transferability, which refers to the ability of adversarial examples generated on surrogate models to deceive other target models, even when the target models have different architectures or are trained on different datasets (Xie et al., 2019b; Wang & He, 2021).

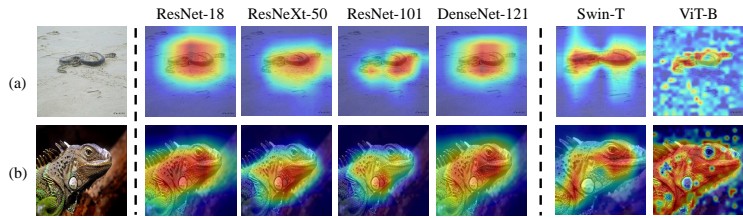

Figure 1: **Demonstration of different discriminative regions of different models.** We adopt Grad-CAM (Selvaraju et al., 2016) to visualize the attention maps of four CNN-based models (ResNet-18, ResNeXt-50, ResNet-101, DenseNet-121) and two Transformer-based models (Swin-T, ViT-B).

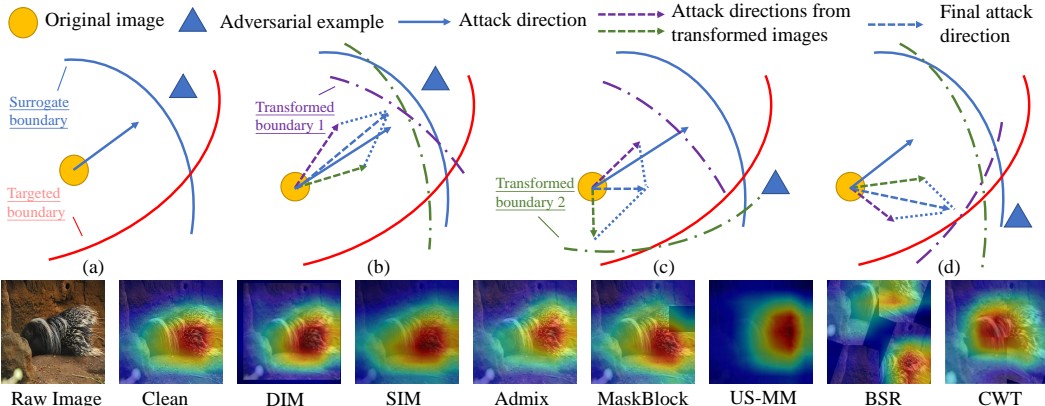

Figure 2: **Top: Overview of different attack schemes. Bottom: Grad-CAM visualizations of model attention on ResNet-18.** **(a)** Standard gradient-based attacks tend to overfit the surrogate model, resulting in an attack direction that fails to cross the target model's decision boundary. **(b)** Some transformation-based methods, such as DIM and SIM, often generate gradients from transformed images that are too similar to the original, thus failing to provide significant directional diversity. **(c)** While advanced methods like BSR can diversify attack directions, some of their gradients may incorrectly focus on irrelevant background regions, which can reduce the attack's transferability. **(d)** In contrast, leveraging its local transformation strategy, CWT generates a diverse set of potent gradients that effectively guide the adversarial example across the target decision boundary.

Researchers have developed various methods to generate adversarial examples and improve adversarial transferability. Gradient-based methods (Dong et al., 2018; Gao et al., 2020) leverage the gradient information of models to craft adversarial examples with high computational efficiency and serve as the foundation for many other approaches. Model-related methods (Wu et al., 2020; Ma et al., 2025) leverage internal model features to create architecture-specific adversarial examples. While effective for their intended architectures, they tend to overfit and lack generalizability. Ensemble-based (Liu et al., 2016; Chen et al., 2023) and generation-based methods (Salzmann et al., 2021) improve transferability by combining multiple model outputs or using generative models like GANs, but at a high computational cost. In contrast, input transformation-based methods (Xie et al., 2019b; Zhang et al., 2023) enhance cross-model transferability by diversifying adversarial examples through operations like scaling and rotation. These methods are computationally efficient and don't require model access, making them suitable for black-box attack scenarios.

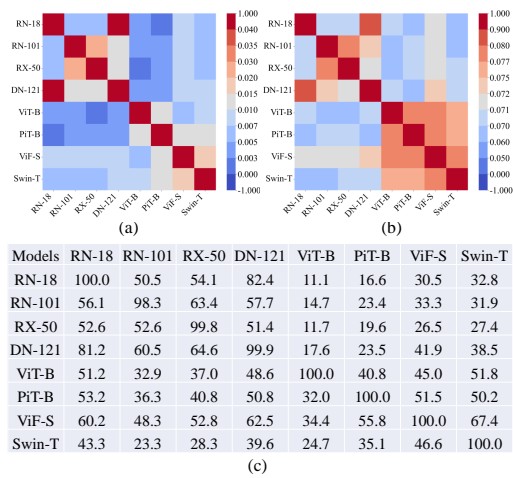

| Models | RN-18 | RN-101 | RX-50 | DN-121 | ViT-B | PiT-B | ViF-S | Swin-T |
|--------|-------|--------|-------|--------|-------|-------|-------|--------|
| RN-18 | 100.0 | 50.5 | 54.1 | 82.4 | 11.1 | 16.6 | 30.5 | 32.8 |
| RN-101 | 56.1 | 98.3 | 63.4 | 57.7 | 14.7 | 23.4 | 33.3 | 31.9 |
| RX-50 | 52.6 | 52.6 | 99.8 | 51.4 | 11.7 | 19.6 | 26.5 | 27.4 |
| DN-121 | 81.2 | 60.5 | 64.6 | 99.9 | 17.6 | 23.5 | 41.9 | 38.5 |
| ViT-B | 51.2 | 32.9 | 37.0 | 48.6 | 100.0 | 40.8 | 45.0 | 51.8 |
| PiT-B | 53.2 | 36.3 | 40.8 | 50.8 | 32.0 | 100.0 | 51.5 | 50.2 |
| ViF-S | 60.2 | 48.3 | 52.8 | 62.5 | 34.4 | 55.8 | 100.0 | 67.4 |
| Swin-T | 43.3 | 23.3 | 28.3 | 39.6 | 24.7 | 35.1 | 46.6 | 100.0 |

(c)

Figure 3: **(a)** Cosine similarity of gradients computed across different models. **(b)** Cosine similarity of the adversarial noise generated using different models as surrogates. **(c)** The corresponding ASR matrix. The attack method is MI-FGSM.

However, existing input transformation-based methods still face critical challenges. As illustrated in Fig. 2, some approaches fail to effectively alter the model's attention, resulting in an attention distribution that remains consistent with the original image. In contrast, others misdirect attention to irrelevant regions of the object, deviating from the areas of interest in the original image.

To address these issues, we propose a novel method called Component-Wise Transformation (CWT), which enhances the transferability of adversarial examples by encouraging models to focus on diverse regions of the object in the original image. Specifically, CWT applies block-wise transformations that interpolate and selectively rotate image patches while preserving essential semantic infor-

mation. By introducing these localized transformations, our method generates adversarial examples with enriched attention distributions that generalize effectively across different model architectures.

Our contributions can be summarized as follows:

- We introduce a novel perspective on addressing the challenge of untargeted adversarial transferability. Instead of broadly enriching input images, our approach focuses on generating transformed images that enable a single surrogate model to emulate the gradient diversity typically achieved by multi-model ensembles.

- We propose CWT, an innovative method combining interpolation and selective rotation to enhance adversarial transferability by diversifying attention distributions.

- Our method achieves state-of-the-art results on the ImageNet and COCO datasets, demonstrating superior attack success rates compared to existing approaches.

## 2 RELATED WORK

### 2.1 INPUT TRANSFORMATION-BASED ATTACKS

Input transformation-based methods have been widely studied to enhance adversarial transferability, particularly in black-box settings. These techniques work by diversifying the input samples used for gradient computation, which helps adversarial examples generalize across various model architectures. Early approaches, such as scale-based methods (Xie et al., 2019b; Lin et al., 2019), primarily focused on resizing and transforming entire input images. Extending this concept, mixed image-based methods (Wang et al., 2021; 2023a) further enriched input diversity by incorporating features from other images. However, transformations applied to the whole image alone may not always effectively alter a model's attention to specific regions. To address this limitation, recent research has introduced block-based methods (Fan et al., 2022; Wang et al., 2024). These methods partition input images into smaller blocks and apply transformations independently to each block. Furthermore, to optimize and better leverage existing transformation techniques, adaptive-based methods (Zhu et al., 2024) introduce learnable transformation strategies to optimize adversarial transferability.

Despite these advancements, existing methods still suffer from insufficient gradient diversity. To overcome this limitation, CWT enables a single surrogate model to emulate the gradient diversity typically achieved by multi-model ensembles. By purposefully designing transformations, CWT effectively directs the surrogate model's attention to diverse object regions.

### 2.2 ADVERSARIAL DEFENSE

Various defense approaches have been proposed to mitigate the threat of adversarial attacks, including adversarial training (Madry, 2017), feature denoising (Xie et al., 2019a), certified defenses (Raghunathan et al., 2018; Gowal et al., 2019) and diffusion purification (Wang et al., 2023b). Among these, AT (Shafahi et al., 2019) is one of the most effective methods, where adversarial examples are incorporated during training to enhance model robustness. HGD (Liao et al., 2018) utilizes a denoising autoencoder guided by high-level representations to eliminate adversarial perturbations. Similarly, NRP (Naseer et al., 2020) leverages a self-supervised adversarial training mechanism to purify input samples, demonstrating strong effectiveness. Certified defense methods, such as RS (Cohen et al., 2019), train robust classifiers by adding noise to the input and providing provable robustness within a certain radius. Furthermore, diffusion purification such as Diffpure (Nie et al., 2022) employs diffusion models to process input samples and reduce adversarial perturbations.

## 3 METHODOLOGY

In this section, we first introduce the prerequisite knowledge and our motivation. Then, we provide a detailed explanation of our CWT method. We also summarize the distinctions between our method and DIM Xie et al. (2019b), as well as BSR Wang et al. (2024) in Appendix A.8.

### 3.1 PRELIMINARIES

Given a target model $f$ with parameters $\theta$ and a clean input $x$ with the ground-truth label $y$, the objective of an untargeted attacker is to generate an adversarial example $x^{\text{adv}}$ that is visually similar to $x$, such that $\|x^{\text{adv}} - x\|_p \leq \epsilon$ and causes the model to misclassify the input, meaning $f(x^{\text{adv}}; \theta) \neq y$. Here, $\epsilon$ denotes the perturbation budget, and $\|\cdot\|_p$ is the $\ell_p$-norm that quantifies the allowed perturbation. In this paper, we focus on the $\ell_\infty$-norm (where $p = \infty$), which restricts the maximum change to each individual pixel. To generate such an adversarial example, the attacker generally maximizes the loss function $J(x^{\text{adv}}, y; \theta)$ (e.g., cross-entropy loss) with respect to $x^{\text{adv}}$, subject to the constraint $\|x^{\text{adv}} - x\|_p \leq \epsilon$. This can be formalized as follows:

$$x^{\text{adv}} = \underset{\|x^{\text{adv}} - x\|_p \leq \epsilon}{\arg\max} \; J(x^{\text{adv}}, y; \theta). \tag{1}$$

Many methods have been proposed to approximate this optimization. Among these, MI-FGSM (Dong et al., 2018) incorporates a momentum term into the gradient calculation to stabilize the perturbation direction across iterations, thereby increasing the attack success rates on various models. The update rule for the accumulated gradient $g_t$ is:

$$g_{t+1} = \mu \cdot g_t + \frac{\nabla_x J(x_t^{\text{adv}}, y; \theta)}{\|\nabla_x J(x_t^{\text{adv}}, y; \theta)\|_1}, \tag{2}$$

$$x_{t+1}^{\text{adv}} = x_t^{\text{adv}} + \alpha \cdot \text{sign}(g_{t+1}). \tag{3}$$

where $\mu$ is the decay factor that controls the influence of previous gradients, and $g_t$ is the accumulated gradient at iteration $t$. The $\ell_1$-normalization of the gradient helps maintain consistency in the perturbation direction across iterations, enhancing the transferability of the attack.

### 3.2 MOTIVATION

Despite the effectiveness of methods like MI-FGSM in stabilizing the attack process, a critical challenge remains: adversarial examples crafted on a surrogate model often fail to transfer to target models, especially those with different architectures. This observation raises a fundamental question: Why does this discrepancy in transferability occur?

For the remainder of this section, we use the subscripts s and t to denote the surrogate and target models, respectively.

**Proposition 1.** *The success of gradient-based transfer attacks is directly proportional to the alignment between the surrogate model's gradient and the target model's gradient.*

*Derivation.* In a standard gradient-based attack, the surrogate perturbation $\delta_s$ is crafted by taking a small step of size $\alpha$ in the direction of the *surrogate* model's gradient:

$$\delta_s = \alpha \cdot \nabla_x J_s(x_s^{\text{adv}}, y; \theta_s) \tag{4}$$

To measure the effect of this perturbation on the target model, we can approximate the change in the target loss, $\Delta J_t$, using a first-order Taylor expansion:

$$\Delta J_t = J_t(x + \delta_s, y; \theta_t) - J_t(x, y; \theta_t) \approx \delta_s^T \nabla_x J_t(x, y; \theta_t) \tag{5}$$

By substituting the definition of $\delta_s$ into Eq. 4, we obtain:

$$\Delta J_t \approx \alpha \cdot (\nabla_x J_s(x_s^{\text{adv}}, y; \theta_s))^T (\nabla_x J_t(x, y; \theta_t)) \tag{6}$$

This formula reveals that the transferability of an attack fundamentally depends on the alignment between the surrogate gradient and the target gradient. We empirically validate this relationship with an experiment on 1000 ImageNet images, as shown in Fig. 3. The results demonstrate that models with more similar architectures tend to have higher cosine similarity in their gradients. This, in turn, leads to the generation of more similar adversarial noise patterns. Consequently, higher similarity in the adversarial noise strongly correlates with higher ASR.

However, in practice, these gradients often misalign. The primary reason for this **gradient mismatch** is that different models learn to rely on different sets of discriminative features to make their

Table 2: Attack success rates (%) on eight models with various single input transformations. The surrogate models are **CNN-based**. * indicates the surrogate model.

| Model | Attack | RN-18 (↑) | RN-101 (↑) | RX-50 (↑) | DN-121 (↑) | ViT-B (↑) | PiT-B (↑) | ViF-S (↑) | Swin-T (↑) |
|-------|--------|-----------|------------|-----------|------------|-----------|-----------|-----------|------------|
| RN-18 | DIM | **100*** | 61.7 | 66.1 | 90.4 | 30.4 | 37.4 | 53.4 | 56.9 |
|       | SIM | **100*** | 59.6 | 64.1 | 90.5 | 24.6 | 35.7 | 49.0 | 53.3 |
|       | Admix | **100*** | 69.9 | 74.6 | 95.4 | 31.2 | 42.6 | 59.8 | 63.0 |
|       | BSR | **100.0*** | 89.1 | 90.2 | **99.4** | 49.1 | 62.3 | 79.4 | 79.2 |
|       | CWT | **100.0*** | **90.2** | **93.7** | **99.4** | **55.9** | **68.8** | **84.1** | **83.6** |
| RN-101 | DIM | 61.7 | 84.7* | 63.0 | 65.5 | 30.2 | 40.9 | 48.0 | 48.2 |
|        | SIM | 62.5 | 91.7* | 64.7 | 66.3 | 25.8 | 38.1 | 46.3 | 46.1 |
|        | Admix | 74.7 | 94.9* | 77.5 | 77.4 | 36.0 | 49.7 | 60.9 | 58.6 |
|        | BSR | 86.7 | 94.6* | 89.6 | 90.2 | 58.5 | 72.9 | 80.8 | 78.1 |
|        | CWT | **87.7** | **95.1*** | **91.0** | **91.6** | **67.3** | **78.5** | **85.5** | **81.9** |
| RX-50 | DIM | 61.3 | 55.8 | 86.7* | 63.0 | 26.0 | 35.2 | 45.7 | 44.4 |
|       | SIM | 59.5 | 57.8 | 94.0* | 64.6 | 20.9 | 32.9 | 39.7 | 41.6 |
|       | Admix | 71.8 | 72.9 | 95.8* | 75.6 | 29.6 | 44.4 | 53.7 | 53.9 |
|       | BSR | 85.7 | 86.0 | **96.5*** | 88.7 | 48.2 | 68.3 | 76.0 | 73.9 |
|       | CWT | **87.3** | **86.4** | 95.9* | **90.2** | **57.7** | **72.3** | **80.9** | **78.4** |

classifications. As illustrated in Fig. 1, even when classification results are identical, the attention distributions of different models vary significantly. For instance, as shown in Fig. 1 (a), ResNet-18 focuses on the belly of the snake, while Swin focuses on both the head and tail of the snake. This analysis reframes our central research question: ***How can we compute a surrogate gradient that better aligns with the gradients of various unknown target models?***

A straightforward approach is to ensemble the gradients from multiple surrogate models $\{f_{s_1}, ..., f_{s_K}\}$. The averaged gradient $\bar{g}_s = \frac{1}{K} \sum_{i=1}^{K} \nabla_x J(x, y; \theta_{s_i})$ provides a more robust estimate, improving the expected alignment: $\Delta J_t \approx \alpha \cdot (\bar{g}_s)^T (\nabla_x J_t)$. While effective, this strategy suffers from high computational costs. An alternative strategy is to enrich the input space of a single surrogate model by computing an expected gradient over a distribution of transformations $\mathcal{T}$. This leads to an update direction $g_s^{\mathcal{T}} = \mathbb{E}_{T \sim \mathcal{T}}[\nabla_x J_s(T(x), y; \theta_s)]$, which similarly aims to improve alignment. Admittedly, existing input transformation-based methods inherently diversify the input. However, because their transformations were not explicitly designed to maximize this gradient diversity, they often fail to meaningfully alter the model's attention, as shown in Fig. 2. For example, methods like DIM, SIM, and Admix produce attention distributions nearly identical to the original, while BSR often shifts focus to irrelevant background areas.

To address this limitation, our work aims to generate transformed images that strategically encourage models to focus on diverse regions of an object. To quantitatively evaluate this capability, we measure the IoU between the model's attention regions and the object's actual foreground. Specifically, we compute the overlap between the aggregated Grad-CAM attention map from $N = 20$ transformed images ($M_{CAM}$, thresholded at 0.6) and the foreground segmented by SAM ($M_{SAM}$) (Kirillov et al., 2023). The IoU is defined as:

Table 1: Average IoU between aggregated attention regions and SAM-segmented foregrounds for different models.

| Method | RN-18 | RN-101 | RX-50 |
|--------|-------|--------|-------|
| BSR | 0.425 | 0.423 | 0.427 |
| CWT | 0.501 | 0.499 | 0.489 |

$$IoU = \frac{M_{CAM} \cap M_{SAM}}{M_{CAM} \cup M_{SAM}}. \tag{7}$$

We conducted this experiment on 400 images from the ImageNet set. As shown in Table 1, our proposed method (CWT) achieves a higher IoU than the previous SOTA, demonstrating its superior ability to direct the model's attention across the entire object of interest.

## 3.3 Component-Wise Transformation

Based on the preceding discussion, our objective is to devise transformations that compel the surrogate model to attend to diverse regions of an object. A naïve approach is random masking, which forces the model to rely on unobscured features for its prediction. However, this method often discards critical information and thus degrades attack efficacy. An alternative, inspired by findings that both humans and models are biased towards larger visual features (Proulx, 2010; Zhang et al., 2025), is to apply a zoom-in operation. The goal is to enlarge certain parts of the object to shift the

Table 3: Attack success rates (%) on eight models with various single input transformations. The surrogate models are **Transformer-based**. * indicates the surrogate model.

| Model | Attack | RN-18 (↑) | RN-101 (↑) | RX-50 (↑) | DN-121 (↑) | ViT-B (↑) | PiT-B (↑) | ViF-S (↑) | Swin-T (↑) |
|---|---|---|---|---|---|---|---|---|---|
| **PiT-B** | DIM | 59.5 | 50.5 | 54.5 | 62.2 | 47.9 | 91.6* | 63.5 | 65.3 |
| | SIM | 58.6 | 45.4 | 48.7 | 60.1 | 38.3 | 97.7* | 56.8 | 60.9 |
| | Admix | 60.4 | 47.7 | 51.8 | 60.4 | 42.9 | 94.6* | 61.1 | 63.5 |
| | BSR | 82.9 | 80.8 | 84.0 | 86.5 | 75.0 | 97.8* | 90.2 | 90.4 |
| | CWT | **85.8** | **85.1** | **87.3** | **90.7** | **85.3** | 97.8* | **92.6** | **92.6** |
| **ViF-S** | DIM | 71.2 | 63.7 | 67.1 | 75.2 | 53.4 | 71.0 | 95.1* | 76.9 |
| | SIM | 68.1 | 60.5 | 62.4 | 72.0 | 49.7 | 65.9 | 96.7* | 74.8 |
| | Admix | 75.1 | 67.0 | 70.6 | 78.3 | 56.5 | 72.8 | 97.0* | 81.6 |
| | BSR | **90.7** | 86.9 | **90.7** | 93.5 | 73.3 | 88.9 | **99.3*** | 92.7 |
| | CWT | 89.1 | **87.6** | 89.9 | **94.0** | **80.4** | **91.9** | 99.1* | **93.7** |
| **Swin-T** | DIM | 67.2 | 53.4 | 56.8 | 68.8 | 48.7 | 65.3 | 69.3 | 96.2* |
| | SIM | 48.9 | 29.9 | 34.1 | 45.4 | 27.7 | 35.5 | 43.1 | 98.1* |
| | Admix | 55.1 | 33.9 | 37.4 | 50.0 | 28.3 | 37.8 | 47.2 | 98.2* |
| | BSR | 88.7 | 82.5 | 86.0 | 91.2 | 71.6 | 89.7 | 91.0 | 98.3* |
| | CWT | **90.7** | **85.9** | **88.1** | **93.6** | **80.8** | **92.9** | **94.1** | 98.5* |

model's attention. However, globally scaling up the image fails to effectively change the relative size differences between different parts of the object and can cause significant displacement of the object from its original position. To overcome these limitations, we propose **CWT**, which applies a sequence of transformations to individual blocks of an image (for a detailed theoretical analysis, see Appendix A.2) These transformations include **bilinear interpolation** (for more details on interpolation, see Appendix A.5), comprising **pre-interpolation** (interpolation-based shrinking) and **block-wise scaling** (interpolation-based enlargement), as well as **selective rotation**. Formally, the CWT process involves the following steps:

**Image Partitioning**: Divide the image $x$ into a grid of $n \times n$ non-overlapping blocks $\{B_{i,j}\}$, where $i, j \in \{1, 2, \ldots, n\}$.

**Pre-Interpolation:** For each block $B_{i,j}$, apply **scaled down** to reduce redundancy and compress features. Specifically, $B_{i,j}$ is scaled by a random factor $s_{i,j}$, where:

$$s_{i,j} \sim \text{Uniform}[s_{\min}, s_{\max}], \tag{8}$$

The scaled-down block $B'_{i,j}$ is calculated as:

$$B'_{i,j} = \text{Interpolate}(B_{i,j}, \text{size} = (H'_{i,j}, W'_{i,j})), \ H'_{i,j} = \lfloor H_{i,j}/s_{i,j} \rfloor, \ W'_{i,j} = \lfloor W_{i,j}/s_{i,j} \rfloor \tag{9}$$

The effectiveness of this step is demonstrated in Table 8, which shows how it contributes to reducing redundancy and improving adversarial robustness.

**Block-wise Scaling:** After pre-interpolation, each block is scaled up using the same scaling factor $s_{i,j}$, focusing attention on specific features. The scaled-up block $\hat{B}_{i,j}$ is computed as:

$$\hat{B}_{i,j} = \text{Interpolate}(B'_{i,j}, \text{size} = (H''_{i,j}, W''_{i,j})), \ H''_{i,j} = \lfloor H_{i,j} \cdot s_{i,j} \rfloor, \ W''_{i,j} = \lfloor W_{i,j} \cdot s_{i,j} \rfloor \tag{10}$$

**Selective Rotation:** To further diversify the spatial representation of features and mitigate significant information loss, selective rotation is applied to a subset of the blocks. From the total number of blocks $n^2$, a random subset of $k$ blocks, identified by indices $\mathcal{R}$, is chosen for rotation. For each selected block $\hat{B}_{i,j} \in \mathcal{R}$, a random rotation angle $r_{i,j}$ is uniformly sampled within the range $(-r_{\max}, r_{\max})$, where $r_{\max}$ defines the allowable rotation range. The rotation is performed around the block's center, and any newly exposed areas due to rotation are filled with zeros to maintain the block's original dimensions.

**Random Cropping and Reconstruction:** After scaling and optional rotation, each block $\hat{B}_{i,j}$ is cropped back to its original size $H_{i,j} \times W_{i,j}$. Finally, all the individually transformed blocks $\tilde{B}_{i,j}$ are reassembled into a single transformed image.

To stabilize gradient calculations, we compute the average gradient over $N$ transformed images:

$$\bar{g} = \frac{1}{N} \sum_{i=1}^{N} \nabla_{x^{\text{adv}}} J(T(x^{\text{adv}}, n, N, s_{\max}, k, r_{\max}), y; \theta). \tag{11}$$

---

**Algorithm 1** Component-Wise Transformation

---

**Require:** A raw example $x$ with ground-truth label $y$, the magnitude of perturbation $\epsilon$, learning rate $\alpha$, decay factor $\mu$, number of iterations $T$, number of transformed images $N$, number of blocks $n$, the maximum scaling factor $s_{\max}$, the maximum rotation angle $r_{\max}$, and the number of rotated blocks $k$.

**Ensure:** An adversarial example $x^{\text{adv}}$.

1: Initialize $x_0^{\text{adv}} = x$, $g_0 = 0$, and $\alpha = \epsilon/T$
2: **for** $t = 1 \rightarrow T$ **do**
3:     Calculate the gradient $\bar{g}$ by Eq. 11.
4:     Update the momentum $g_t$ by:
$$g_t = \mu \cdot g_{t-1} + \frac{\bar{g}}{\|\bar{g}\|_1}.$$
5:     Update the adversarial perturbation for this step $\delta$ by:
$$\delta = \alpha \cdot \text{sign}(g_t).$$
6:     Update the adversarial example by adding the perturbation:
$$x_t^{\text{adv}} = x_{t-1}^{\text{adv}} + \delta.$$
7:     Project $x_t^{\text{adv}}$ to ensure it is in the $\epsilon$-ball of $x$:
$$x_t^{\text{adv}} = x + \text{clip}(x_t^{\text{adv}} - x, -\epsilon, \epsilon).$$
8: **end for**
9: **return** $x_T^{\text{adv}}$

---

Here we integrate our CWT method into MI-FGSM, and summarize the algorithm in Algorithm 1.

## 4 EXPERIMENTS

### 4.1 EXPERIMENTAL SETUP

**Dataset.** Following previous works (Wang et al., 2024), we evaluate our proposed CWT on 1000 images belonging to 1000 categories from the validation set of ImageNet dataset. Additionally, we randomly sampled 1,000 images from the COCO (Lin et al., 2014) dataset for further testing.

**Models.** We evaluate our proposed CWT method across three categories of target models: 1) **CNN-based models**, including four widely recognized architectures: ResNet-18 (RN-18) (He et al., 2016), ResNet-101 (RN-101) (He et al., 2016), ResNeXt-50 (RX-50) (Xie et al., 2017), and DenseNet-121 (DN-121) (Huang et al., 2017); 2) **Transformer-based models**, comprising ViT-B (Dosovitskiy, 2020), PiT-B (Heo et al., 2021), Visformer (ViF-S) (Chen et al., 2021), and Swin-Tiny (Swin-T) (Liu et al., 2021); 3) **Defense models**, which include four defense methods: AT (Shafahi et al., 2019), HGD (Liao et al., 2018), RS (Cohen et al., 2019), and DiffPure (Nie et al., 2022); All models are pre-trained on the ImageNet dataset and evaluated on single model.

**Baselines.** We compare CWT with other input transformation-based methods. Specifically, the scale-based methods (DIM (Xie et al., 2019b), SIM (Lin et al., 2019)), the mixed image-based methods (Admix (Wang et al., 2021)), and the block-based methods (BSR (Wang et al., 2024)). For fairness, all the input transformations are integrated into MI-FGSM (Dong et al., 2018). Further baseline comparisons are detailed in Appendix A.3.

**Evaluation Settings.** We set the maximum perturbation $\epsilon = 16/255$, the number of iterations $epoch = 10$, the step size $\alpha = \epsilon/epoch$, the $batchsize = 8$, and the decay factor $\mu = 1$ for MI-FGSM. For our method, CWT generates 20 scaled copies per iteration, divides the image into 2x2 blocks, applies a scaling factor ranging from 1.0 to 1.3, and applies $r_{\max} = 26°$, selectively rotating $k = 2$ blocks. For other methods, we follow the parameters reported in the original papers.

### 4.2 EVALUATIONS ON CNN AND TRANSFORMER-BASED MODELS

We evaluate the transferability of adversarial examples using surrogate models based on both CNN and Transformer architectures, with results summarized in Tab. 2 and Tab. 3, respectively, where the first column indicates the surrogate model, the second column lists the attack methods, and the

Table 4: Attack success rates (%) of adversarial examples generated using various attack methods under different **defense methods**. Configurations are provided in Appendix A.7.

| Defense | Attack | RN-18 (↑) | RN-101 (↑) | RX-50 (↑) | DN-121 (↑) | ViT-B (↑) | PiT-B (↑) | ViF-S (↑) | Swin-T (↑) |
|---|---|---|---|---|---|---|---|---|---|
| AT | DIM | 36.4 | 32.9 | 32.0 | 34.6 | 33.4 | 33.2 | 33.5 | 33.7 |
| | SIM | 36.5 | 31.6 | 32.1 | 35.6 | 35.4 | 33.4 | 34.3 | 31.0 |
| | Admix | 38.4 | 32.6 | 32.2 | 38.0 | 35.9 | 33.5 | 34.3 | 31.8 |
| | BSR | 40.3 | 36.9 | 35.2 | 38.9 | **38.0** | 35.1 | 36.7 | 37.5 |
| | CWT | **41.1** | **38.0** | **36.9** | **39.9** | 37.7 | **36.9** | **37.7** | **39.4** |
| Diffpure | DIM | 24.4 | 29.8 | 20.7 | 27.2 | 25.7 | 23.4 | 26.8 | 21.1 |
| | SIM | 23.3 | 28.9 | 18.0 | 24.7 | 27.8 | 20.9 | 25.9 | 14.2 |
| | Admix | 24.2 | 35.2 | 23.3 | 30.5 | 28.9 | 23.1 | 29.3 | 14.9 |
| | BSR | 33.6 | 45.5 | 28.8 | 33.8 | 39.5 | 35.0 | 37.9 | 30.5 |
| | CWT | **40.5** | **56.7** | **38.4** | **41.3** | **43.2** | **43.2** | **48.7** | **37.8** |
| HGD | DIM | 56.9 | 49.7 | 42.1 | 66.7 | 43.0 | 45.7 | 57.2 | 46.2 |
| | SIM | 52.7 | 43.5 | 36.4 | 64.3 | 42.4 | 37.2 | 49.1 | 22.5 |
| | Admix | 63.1 | 58.2 | 50.6 | 76.5 | 46.8 | 39.9 | 57.2 | 25.9 |
| | BSR | 86.9 | 80.9 | 74.9 | 90.1 | 71.9 | 74.2 | 81.0 | 75.0 |
| | CWT | **91.2** | **85.4** | **81.9** | **93.1** | **73.9** | **81.8** | **85.0** | **83.5** |
| RS | DIM | 26.0 | 21.9 | 22.0 | 25.2 | 23.6 | 22.2 | 22.9 | 22.9 |
| | SIM | 26.0 | 21.3 | 21.3 | 26.0 | 25.2 | 22.7 | 24.0 | 21.5 |
| | Admix | 27.9 | 22.7 | 22.5 | 27.8 | 26.0 | 22.6 | 24.2 | 21.1 |
| | BSR | 27.8 | 26.2 | 25.1 | 26.7 | 27.3 | 26.0 | 26.1 | 24.9 |
| | CWT | **30.9** | **28.7** | **27.9** | **30.8** | **29.3** | **27.8** | **29.5** | **28.7** |

remaining columns present the ASR of different classification models under attack. Our proposed method consistently achieves SOTA performance across almost all experimental settings. Specifically, when the surrogate model is RN-18, our method achieves the highest ASR across all target models. Furthermore, when targeting ViT-B and PiT-B, it surpasses the previous SOTA BSR by more than 5% and outperforms other baseline methods by at least 17.4%. Notably, while CWT provides limited improvement over BSR when the surrogate is ViF-S—a model with a global attention mechanism that already results in dispersed attention—it achieves significant gains with local/window-based attention models like PiT-B and Swin-T by better directing focus to diverse image regions, as shown in Fig. 1.

## 4.3 EVALUATIONS ON DEFENSE METHOD

To comprehensively assess the robustness of our proposed method, we conducted experiments under four defense settings: Adversarial Training (AT), High-level representation Guided Denoiser (HGD), Randomized Smoothing (RS), and Diffusion Purification (Diffpure). More extensive evaluations can be found in Appendix A.4. As summarized in Tab. 4, our method CWT consistently achieves the highest ASR across most of defense settings. For instance, under Diffpure, it outperforms BSR by 8.1% in mean ASR and exceeds weaker baselines by more than 17%.

## 4.4 EVALUATIONS ON MORE DATASETS (COCO)

To further evaluate adversarial transferability, we sampled 1,000 images from the COCO dataset that were unanimously classified by all surrogate models. We then compared the CWT and BSR attacks on various Transformer-based architectures, using a CNN-based model as the surrogate. As shown in Tab. 5, the results demonstrate that CWT consistently outperforms BSR.

## 4.5 PERFORMANCE VS. RUNTIME

As shown in Tab. 6, We benchmark CWT against BSR on an A100 GPU (batch size 20, RN-18 surrogate). Across all configurations, CWT introduced only a minimal 10ms per image overhead compared to BSR, highlighting that computational efficiency is predominantly influenced by the underlying framework rather than the algorithmic modifications.

## 4.6 ABLATION STUDY

To further evaluate CWT, we conduct ablation studies on five key hyperparameters and one critical step: *number of blocks $n$, maximum scale factor $s_{max}$, maximum rotation angle $r_{max}$, number of rotated blocks $k$, number of transformed copies $N$*, and the key step *pre-interpolation*.

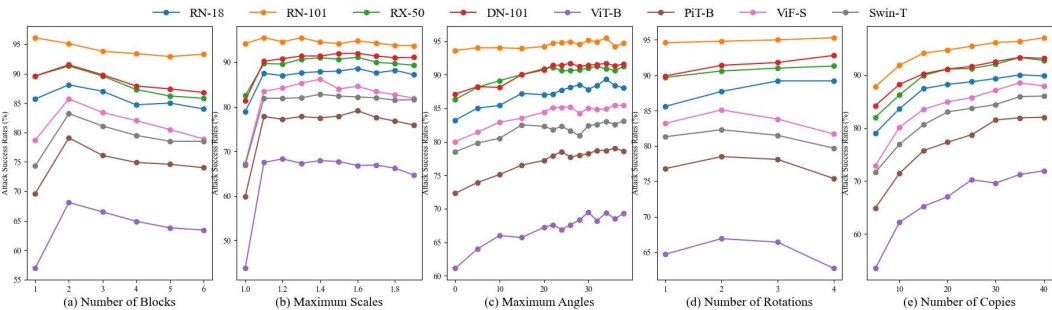

Figure 4: Attack success rates (%) of various models on the adversarial examples generated by CWT with different numbers of blocks, maximum resize rates, maximum rotate angles, numbers of rotation blocks and numbers of transformed copies. The adversarial examples are crafted using the RN-101 model and tested on seven other models under the black-box setting.

Table 5: ASR (%) on COCO Dataset.

| Model | Attack | ViT-B | PiT-B | ViF-S | Swin-T |
|---|---|---|---|---|---|
| **RN-18** | BSR | 48.8 | 65.5 | 82.8 | 78.7 |
| | CWT | **55.8** | **71.9** | **86.6** | **83.8** |
| **RN-101** | BSR | 62.9 | 81.2 | 86.6 | 82.3 |
| | CWT | **71.1** | **84.6** | **88.7** | **86.3** |
| **RX-50** | BSR | 51.7 | 75.2 | 82.1 | 77.0 |
| | CWT | **60.6** | **79.6** | **86.3** | **82.9** |

Table 6: Efficiency under RN-18.

| Copies | Attack | DN-121 | ViT-B | PiT-B | Time |
|---|---|---|---|---|---|
| 10 | BSR | 98.5 | 45.6 | 57.7 | 37.89 ms |
| | CWT | 98.8 | 50.8 | 63.8 | 45.21 ms |
| 15 | BSR | 98.9 | 48.9 | 62.8 | 51.57 ms |
| | CWT | 99.3 | 53.4 | 66.5 | 65.19 ms |
| 20 | BSR | 99.4 | 49.1 | 62.3 | 73.30 ms |
| | CWT | 99.4 | 55.9 | 68.8 | 81.99 ms |

As shown in Figure 4, when $n > 1$, $s_{max} > 1.1$, $r_{max} > 20°$, and $k > 0$, the models generally maintain stable performance, achieving optimal results at $n = 2$, $s_{max} = 1.3$, $r_{max} = 26°$, and $k = 2$. We also present the mean and standard deviation from multiple trials with randomly selected values beyond the stability thresholds in Tab. 7. Notably, even our worst-case performance surpasses BSR's results. For the number of transformed copies ($N$), the ASR steadily improves with increasing $N$, as this stabilizes gradient updates and introduces greater diversity. However, to maintain consistency with the BSR baseline and minimize computational overhead, we set $N = 20$ in our experiments.

Table 7: Hyperparameter Robustness under RN-101.

| Attack | RN-18 | RX-50 | PiT-B | ViF-S | Swin-T |
|---|---|---|---|---|---|
| BSR | 86.7 | 89.6 | 72.9 | 80.8 | 78.1 |
| CWT | **88.4**±0.3 | **90.7**±0.3 | **77.1**±0.7 | **83.9**±0.6 | **81.8**±0.3 |

Table 8: Ablation Study on CWT Components with PiT-B Surrogate. 'In' and 'Ro' denote Pre-interpolation and Rotation, respectively.

| Operation | | RN-18 | RN-101 | RX-50 | DN-121 | ViT-B | ViF-S | Swin-T |
|---|---|---|---|---|---|---|---|---|
| In | Ro | | | | Model: PiT-B | | | |
| ✓ | ✓ | **85.8** | **85.1** | **87.3** | **90.7** | **85.3** | **92.6** | **93.7** |
| ✓ | | 83.9 | 82.6 | 85.3 | 87.3 | 82.0 | 91.5 | 90.9 |
| | ✓ | 79.3 | 79.0 | 80.2 | 84.5 | 80.0 | 88.1 | 89.1 |
| | | 76.3 | 73.4 | 75.7 | 80.4 | 70.4 | 84.8 | 85.9 |

To further dissect the role of each component, we conducted additional ablation studies on pre-interpolation and rotation under PiT-B, as shown in Tab. 8. The results reveal that pre-interpolation offers a more substantial contribution to performance than rotation, though optimal efficacy is achieved when both components are used in conjunction.

## 5 CONCLUSION

Based on the observation that inconsistent attention distributions across models are a major barrier to adversarial example transferability, we propose CWT, a method utilizing interpolation and selective rotation. CWT effectively generates diverse transformed images, enabling a single surrogate model to focus on different regions of the target object. Experimental results demonstrate that our approach surpasses existing SOTA methods in ASR, highlighting its generalization capabilities. Furthermore, CWT offers a novel perspective for input transformation-based adversarial attacks.

ETHICS STATEMENT

A potential negative impact of our work is that malicious actors could employ our method to attack deployed models. However, the motivation for this research is fundamentally defensive. The core contribution of this paper is to demonstrate a critical vulnerability in deep learning models: adversarial attacks can be made significantly more transferable by manipulating inputs to diversify the model's regions of attention. By publishing these findings, we aim to equip the security community with a deeper understanding of this threat vector. In conclusion, our work details this potent attack algorithm to emphasize the importance of developing more robust defenses capable of mitigating such attacks

REPRODUCIBILITY STATEMENT

To ensure reproducibility, we provide the complete source code for our method in the supplementary materials. This includes the core implementation of CWT algorithm and scripts to generate data. Detailed experimental settings, including hyperparameters for all methods, model architectures, and evaluation protocols, are described in Section 4.1. Our experiments utilize the public ImageNet and COCO datasets; the code for the specific sampling procedure used for the COCO dataset is also included in the supplementary materials. Finally, the theoretical motivation and derivations for our approach are provided in Section 3.2.

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

# A APPENDIX

## A.1 USE OF LARGE LANGUAGE MODELS

During the preparation of this manuscript, Google's Gemini [1], was used as a writing-assistance tool. The use of the LLM was strictly limited to language polishing, which included correcting grammatical errors, rephrasing sentences, and improving overall readability. The LLM did not contribute to any of the core academic components of this work, including research ideation, algorithm design, and experimental methodology. The authors take full responsibility for all content and conclusions presented in this paper.

## A.2 THEORETICAL RATIONALE OF THE CWT METHOD

This section provides a theoretical framework to justify the effectiveness of the CWT. We formalize the intuition that CWT systematically guides a surrogate model's attention to explore diverse, vulnerable regions of a target object. We provide a step-by-step rationale for each of CWT's components.

### A.2.1 FORMALIZATION PRELIMINARIES AND NOTATION

**Definition A.1** (Basic Notations). *Let the following be defined:*

- $x \in \mathbb{R}^{H \times W \times C}$*: The original input image.*

- *$y$: The ground-truth label for $x$.*

- *$f(\cdot; \theta_s)$: A surrogate model parameterized by $\theta_s$.*

- *$J(x, y; \theta_s)$: The loss function evaluated on the surrogate model.*

- *$M_S$: A binary mask for the object region in $x$, where for a pixel coordinate $p$, $M_S(p) = 1$ if $p$ is part of the object, and $M_S(p) = 0$ otherwise. We denote its support as $\mathrm{supp}(M_S)$.*

- *$M_a(x, \theta_s)$: The attention mask (e.g., a Grad-CAM heatmap normalized to $[0, 1]$) of the model for a given input $x$.*

- *$T$: A stochastic transformation operator drawn from the CWT distribution, such that $x' = T(x)$ is a transformed image.*

- *$g(T(x), \theta_s) = \nabla_x J(T(x), y; \theta_s)$: The gradient of the loss with respect to the **original image** $x$, computed on the transformed image $T(x)$.*

**Definition A.2** (High-Attention Region). *For a dense attention mask $M_a$, we define the **High-Attention Region** using a threshold $\tau \in (0, 1)$. This thresholded support, denoted as $\mathrm{supp}_\tau(M_a)$, is the set of pixel coordinates where the attention value exceeds $\tau$:*

$$\mathrm{supp}_\tau(M_a(x, \theta_s)) = \{p \mid M_a(p; x, \theta_s) > \tau\} \tag{12}$$

*The model's effective region of focus is thus $s = \mathrm{supp}_\tau(M_a(x, \theta_s))$.*

### A.2.2 CORE THESIS: UNION COVERAGE OF HIGH-ATTENTION REGIONS

The standard gradient $g(x, \theta_s)$ is overfitted to the initial high-attention region $s$. The CWT algorithm's success stems from its ability to generate a diverse set of transformations that collectively force the model to attend to a much broader portion of the object.

**Proposition A.1** (CWT Optimization Goal: Union Coverage). *The objective of CWT is to ensure that a set of its transformations $\{T_i\}_{i=1}^N$ collectively expands the high-attention region within the object's boundary. Formally:*

$$\mathrm{Area}\left(\left(\bigcup_{i=1}^N \mathrm{supp}_\tau(M_a(T_i(x), \theta_s))\right) \cap \mathrm{supp}(M_S)\right) > \mathrm{Area}\left(\mathrm{supp}_\tau(M_a(x, \theta_s)) \cap \mathrm{supp}(M_S)\right) \tag{13}$$

---

[1] https://gemini.google.com/

### A.2.3 RATIONALE OF CWT COMPONENTS

**Lemma A.1** (Feature Scale Preference). *For a DNN pre-trained on natural images, a semantic feature presented at a larger physical scale in the input will elicit a stronger neuronal activation response. Let $A_k(x)$ be the feature activation map of the $k$-th layer for input $x$. If images $x_1$ and $x_2$ contain the same object, but the object in $x_2$ is larger than in $x_1$, it is highly probable that $\|A_k(x_2)\|_F > \|A_k(x_1)\|_F$, where $\|\cdot\|_F$ is the Frobenius norm. This stronger activation cascade leads to higher values in the final attention mask $M_a$.*

**The Role of Pre-Interpolation: Preparing for Effective Scaling**  Pre-Interpolation, via bilinear down-sampling ($D_s$), acts as a low-pass filter to reduce informational redundancy (e.g., high-frequency textures). Let $\mathcal{F}(B)$ be the Fourier transform of a block $B$, and let $\mathcal{H}(\cdot)$ be an operator measuring energy in high-frequency bands. This operation ensures $\mathcal{H}(\mathcal{F}(D_s(B))) < \mathcal{H}(\mathcal{F}(B))$. By attenuating this content, the subsequent up-sampling $U_s$ interpolates based on the block's core structural information. This biases the model to focus on more robust and transferable shape features, rather than surrogate-specific textures.

**The Role of Block-wise Scaling: Actively Guiding Attention**  By scaling up the pre-processed block, this step leverages Lemma A.1 to ensure the transformed region is included in the high-attention support. The transformation $T_{i,j}(B_{i,j}) = U_s(D_s(B_{i,j}))$ enlarges the content within block $B_{i,j}$. This leads to stronger activations and thus higher values in the attention mask $M_a$. The increased attention values make it highly probable that this region will surpass the threshold $\tau$, thus actively guiding it into the high-attention support $\text{supp}_\tau$. By stochastically applying this to different blocks, CWT ensures the terms in the union $\bigcup_{i=1}^{N} \text{supp}_\tau(\dots)$ cover diverse parts of the object.

**The Role of Rotation: Breaking Canonical View Dependencies**  Rotation diversifies the feature basis for gradient computation by breaking the model's reliance on orientation-dependent features. CNNs are not rotation-invariant; their learned filters, $f \in \mathcal{F}$, are tuned to detect patterns in specific orientations. For a rotation operator $R_\alpha$ and a convolution operation $*$, we have:

$$f * R_\alpha(B) \neq R_\alpha(f * B) \tag{14}$$

Applying a rotation forces the model to leverage more abstract and robust features. The resulting high-attention region is thus based on features that are less dependent on the surrogate model's specific orientation biases, making the corresponding gradient more generalizable.

### A.3 EXTENDED EVALUATIONS ON BASELINE METHODS

In this section, we provide an extended comparison of our CWT method against several strong baselines: Maskblock (Fan et al., 2022), US-MM (Wang et al., 2023a), and OPS (Guo et al., 2025). Using RN-18 as the surrogate model, we evaluate the ASR on seven target models and the average runtime per sample on an A100 GPU (batch size = 20), with results detailed in Tab. A1. The results clearly indicate that CWT significantly outperforms Maskblock and US-MM in attack performance. More importantly, CWT demonstrates a superior trade-off between performance and efficiency when compared to the highly competitive OPS method. Specifically, at a comparable runtime to OPS(10, 4, 4), CWT achieves a 6.7% higher average black-box ASR (82.2% vs. 75.5%). Furthermore, even when compared to the strongest but most computationally expensive configuration, OPS(10, 5, 5), our method still holds a 1.7% ASR advantage while being significantly faster (81.99ms vs. 128.73ms).

Table A1: Attack success rates (%) and average runtime per sample on on eight models with various single input transformations. The surrogate model is **RN-18**. * indicates the surrogate model.

| Method | RN-18 (↑) | RN-101 (↑) | RX-50 (↑) | DN-121 (↑) | ViT-B (↑) | PiT-B (↑) | ViF-S (↑) | Swin-T (↑) | Time |
|--------|-----------|------------|-----------|------------|-----------|-----------|-----------|------------|------|
| MaskBlock | 100* | 48.8 | 50.4 | 79.6 | 18.6 | 25.3 | 38.4 | 43.7 | 45.81ms |
| US-MM | 100.0* | 66.6 | 71.7 | 94.3 | 29.1 | 41.0 | 56.8 | 60.3 | 46.40ms |
| OPS(10, 3, 3) | 99.6* | 77.0 | 79.4 | 96.0 | 41.1 | 50.6 | 66.8 | 70.5 | 61.62ms |
| OPS(10, 4, 4) | 99.8* | 82.6 | 85.3 | 97.9 | 50.8 | 60.4 | 75.1 | 76.7 | 87.56ms |
| OPS(10, 5, 5) | 100* | 87.5 | 89.7 | 98.7 | 58.5 | 66.6 | 80.7 | 82.1 | 128.73ms |
| CWT | 100.0* | 90.2 | 93.7 | 99.4 | 55.9 | 68.8 | 84.1 | 83.6 | 81.99ms |

### A.4 EXTENDED EVALUATIONS ON DEFENSE METHODS

In this section, we evaluate the performance of our proposed method on five adversarially trained models, namely Inc-v3$_{ens}$, Inc-v3$_{ens3}$, Inc-v3$_{ens4}$, IncRes-v2$_{ens}$ and Res50$_{in}$ (Tramèr et al., 2017; Geirhos et al., 2018), as well as the NRP (Naseer et al., 2020) method. The results are summarized in Tab. A2 through A4.

Specifically, in Tab. A2, we analyze the attack effectiveness of adversarial samples generated using different attack methods on various surrogate models against adversarially trained models. It is worth noting that these models do not achieve 100% classification accuracy on clean images. However, the misclassification rates of the models are all below 5%. The results indicate that our method surpasses previous SOTA approaches by an average improvement of 5.3%.

For Tab. A3 and A4, we apply NRP to process adversarial images generated by different defense methods and ASR on eight models. The results demonstrate that our method, CWT, outperforms existing techniques in terms of both ASR and standard deviation for CNN-based and Transformer-based surrogate models. Notably, while maintaining a reduced standard deviation, our approach consistently improves the ASR by an average of more than 5% over prior SOTA methods.

Table A2: Attack success rates (%) of adversarial examples generated using various attack methods across eight classification models under **adversarially trained models**.

| Model | Attack | RN-18 (↑) | RN-101 (↑) | RX-50 (↑) | DN-121 (↑) | ViT-B (↑) | PiT-B (↑) | ViF-S (↑) | Swin-T (↑) | Mean (↑) | Std. Dev. (↓) |
|---|---|---|---|---|---|---|---|---|---|---|---|
| Inc-v3$_{ens}$ | DIM | 64.2 | 50.7 | 46.2 | 68.0 | 47.8 | 48.8 | 61.4 | 52.4 | 54.9 | 7.8 |
| | SIM | 60.1 | 45.2 | 41.2 | 67.2 | 49.0 | 44.1 | 54.8 | 28.6 | 48.8 | 11.2 |
| | Admix | 71.6 | 58.9 | 50.6 | 77.9 | 51.4 | 43.8 | 59.5 | 31.1 | 55.6 | 14.0 |
| | MaskBlock | 44.4 | 36.3 | 31.1 | 51.0 | 39.2 | 41.8 | 42.3 | 25.9 | 39.0 | 7.4 |
| | US-MM | 68.7 | 60.1 | 52.8 | 74.4 | 53.5 | 47.2 | 64.8 | 30.8 | 56.5 | 12.9 |
| | BSR | 88.9 | 79.1 | 71.5 | 89.8 | 72.7 | 75.0 | 81.4 | 75.8 | 79.3 | 6.5 |
| | **CWT** | **92.7** | **84.6** | **79.0** | **92.6** | **74.0** | **82.1** | **86.1** | **84.0** | **84.4** | **5.9** |
| Inc-v3$_{ens3}$ | DIM | 60.2 | 48.3 | 43.7 | 65.7 | 44.7 | 47.1 | 58.4 | 49.8 | 52.2 | 7.6 |
| | SIM | 57.1 | 43.7 | 36.4 | 63.8 | 46.7 | 41.2 | 51.7 | 26.1 | 45.8 | 11.1 |
| | Admix | 67.9 | 56.4 | 49.7 | 75.1 | 49.5 | 42.1 | 59.2 | 29.3 | 53.6 | 13.5 |
| | MaskBlock | 43.4 | 36.2 | 28.5 | 51.1 | 36.5 | 38.4 | 40.9 | 24.8 | 37.5 | 7.7 |
| | US-MM | 65.0 | 60.1 | 51.2 | 72.3 | 50.1 | 45.2 | 61.9 | 29.8 | 54.5 | 12.5 |
| | BSR | 85.7 | 77.7 | 70.5 | 87.1 | 71.0 | 71.6 | 78.5 | 73.6 | 77.0 | 6.1 |
| | **CWT** | **91.4** | **82.5** | **77.5** | **90.8** | **72.5** | **79.4** | **85.5** | **81.6** | **82.7** | **6.0** |
| Inc-v3$_{ens4}$ | DIM | 61.5 | 45.2 | 39.1 | 65.2 | 43.9 | 46.0 | 54.8 | 48.7 | 50.6 | 8.5 |
| | SIM | 52.7 | 39.8 | 34.4 | 62.8 | 44.5 | 38.2 | 48.6 | 25.8 | 43.4 | 10.7 |
| | Admix | 64.3 | 53.2 | 45.7 | 72.9 | 47.5 | 40.0 | 54.6 | 28.0 | 50.8 | 13.1 |
| | MaskBlock | 39.1 | 32.3 | 27.8 | 47.0 | 36.8 | 33.7 | 39.3 | 24.9 | 35.1 | 6.6 |
| | US-MM | 61.6 | 56.8 | 46.8 | 71.0 | 43.5 | 43.5 | 58.5 | 27.5 | 51.9 | 12.4 |
| | BSR | 85.1 | 73.6 | 66.8 | 86.4 | 69.2 | 70.5 | 77.2 | 72.9 | 75.2 | 6.8 |
| | **CWT** | **90.9** | **82.0** | **75.4** | **90.8** | **72.1** | **79.1** | **81.7** | **79.9** | **81.5** | **6.2** |
| IncRes-v2$_{ens}$ | DIM | 48.0 | 43.9 | 34.8 | 55.1 | 41.5 | 39.8 | 50.2 | 39.8 | 44.1 | 6.2 |
| | SIM | 42.1 | 36.7 | 30.3 | 50.6 | 39.7 | 33.6 | 44.6 | 19.2 | 37.1 | 9.0 |
| | Admix | 50.6 | 50.1 | 41.6 | 61.8 | 44.9 | 34.2 | 50.6 | 21.0 | 44.4 | 11.6 |
| | MaskBlock | 30.1 | 28.4 | 23.3 | 37.7 | 33.6 | 29.2 | 30.0 | 17.5 | 28.7 | 5.7 |
| | US-MM | 49.3 | 51.7 | 44.0 | 59.1 | 44.4 | 38.4 | 53.2 | 21.0 | 45.1 | 10.9 |
| | BSR | 75.0 | 71.4 | 63.9 | 77.8 | 67.4 | 66.1 | 69.5 | 64.0 | 69.4 | 4.8 |
| | **CWT** | **81.8** | **79.6** | **73.5** | **84.6** | **69.6** | **73.8** | **76.9** | **73.9** | **76.7** | **4.7** |
| Res50$_{in}$ | DIM | 83.8 | 54.1 | 51.1 | 79.6 | 49.7 | 52.9 | 61.8 | 57.6 | 61.3 | 12.3 |
| | SIM | 81.6 | 53.9 | 49.3 | 79.1 | 54.9 | 46.8 | 57.6 | 36.2 | 57.4 | 14.6 |
| | Admix | 90.1 | 67.1 | 61.3 | 87.7 | 58.2 | 49.2 | 64.6 | 40.3 | 64.8 | 16.1 |
| | MaskBlock | 72.4 | 47.1 | 39.8 | 70.5 | 48.3 | 44.7 | 47.6 | 33.3 | 50.5 | 13.0 |
| | US-MM | 88.2 | 73.1 | 66.8 | 87.7 | 52.6 | 70.7 | 70.7 | 41.4 | 67.7 | 15.1 |
| | BSR | 97.1 | 83.2 | 78.8 | 95.3 | **75.1** | 77.0 | 82.3 | 82.3 | 83.9 | 7.6 |
| | **CWT** | **98.1** | **85.7** | **82.2** | **95.3** | 74.6 | **80.5** | **85.7** | **85.9** | **86.0** | **7.1** |

### A.5 MORE ANALYSIS ON INTERPOLATION

To better understand the role of interpolation in affecting model attention distributions, we conducted experiments using various interpolation methods (e.g., bilinear, bicubic, nearest neighbor, and area interpolation) under different scaling factors. As shown in Fig. A1, scaling factors below 1.0 tend to disperse attention across irrelevant regions, reducing focus on critical object areas. Conversely, scaling factors significantly above 1.0 excessively concentrate attention on limited regions. Moderate scaling factors, typically ranging from 1.0 to 1.8, yield the most balanced attention distributions, effectively redistributing focus across diverse object regions. For example, as depicted in Fig. A1 (a), a scaling factor of 1.0 primarily directs attention to a person's back, while a factor of 1.4 shifts focus towards the hips, and 1.8 further moves attention to the left shoulder.

In our method, we strategically apply a two-step process to achieve balanced attention redistribution. First, we perform a shrinking operation on each block to disperse attention and eliminate redundant information. Next, we enlarge the block, refocusing the previously dispersed attention on critical

Table A3: Attack success rates (%) of adversarial examples generated using various attack methods across eight classification models under **NRP**. The surrogate models are **CNN-based.** * indicates the surrogate model.

| Model | Attack | RN-18 (↑) | RN-101 (↑) | RX-50 (↑) | DN-121 (↑) | ViT-B (↑) | PiT-B (↑) | ViF-S (↑) | Swin-T (↑) | Mean (↑) | Std. Dev. (↓) |
|---|---|---|---|---|---|---|---|---|---|---|---|
| **RN-18** | DIM | 98.9* | 39.5 | 40.7 | 64.5 | 19.5 | 25.8 | 30.4 | 34.7 | 44.3 | 25.8 |
| | SIM | 99.5* | 40.6 | 44.7 | 67.5 | 17.2 | 24.9 | 31.9 | 37.3 | 45.5 | 26.5 |
| | Admix | **99.9*** | 47.8 | 51.9 | 77.7 | 19.9 | 28.4 | 35.8 | 41.5 | 50.4 | 26.5 |
| | MaskBlock | 98.0* | 32 | 35.5 | 57.5 | 13.6 | 19 | 25 | 28.1 | 38.6 | 27.4 |
| | US-MM | 99.4* | 46.6 | 51.3 | 76.2 | 21.5 | 29.3 | 39.5 | 42.6 | 50.8 | 25.5 |
| | BSR | 98.8* | 53.4 | 60.0 | 81.8 | 25.9 | 37.7 | 46.7 | 50.0 | 56.8 | 23.5 |
| | CWT | 99.4* | **58.6** | **64.9** | **87.4** | **33.5** | **43.7** | **52.4** | **55.2** | **61.9** | **21.9** |
| **RN-101** | DIM | 54.4 | 56.8* | 38.7 | 49.5 | 19.1 | 25 | 28.5 | 29.4 | 37.7 | 14.4 |
| | SIM | 55.7 | 62.6* | 40.8 | 51 | 17.1 | 25.7 | 28.3 | 30.6 | 39.0 | 16.1 |
| | Admix | 60.9 | 75.7* | 52.9 | 58.9 | 22.6 | 33.2 | 37.9 | 39.8 | 47.7 | 17.4 |
| | MaskBlock | 53.3 | 54.6* | 35.6 | 47.3 | 13 | 19.2 | 23 | 24.9 | 33.9 | 16.2 |
| | US-MM | 99.4* | 46.6 | 51.3 | 76.2 | 21.5 | 29.3 | 39.5 | 42.6 | 50.8 | 25.5 |
| | BSR | 71.9 | 73.4* | 63.6 | 70.8 | 32.2 | 45.0 | 51.8 | 50.1 | 57.7 | 14.9 |
| | CWT | **74.7** | **80.9*** | **68.9** | **75.3** | **40.4** | **52.4** | **58.2** | **57.3** | **63.5** | **13.7** |
| **RX-50** | DIM | 54 | 32.8 | 57.4* | 46.8 | 16 | 23 | 26.8 | 27.2 | 35.5 | 15.3 |
| | SIM | 55.9 | 35.5 | 63.3* | 48.3 | 14.9 | 22.6 | 25.5 | 28.1 | 36.8 | 17.3 |
| | Admix | 60.1 | 45.4 | 76.2* | 56.6 | 19.8 | 29.7 | 34.1 | 35.9 | 44.7 | 18.5 |
| | MaskBlock | 53.4 | 29.3 | 55.2* | 41.9 | 11.9 | 18.4 | 22.6 | 23.7 | 32.1 | 16.2 |
| | US-MM | 68.9 | 49.0 | 78.8* | 63.0 | 31.3 | 37.1 | 39.9 | 48.6 | 48.7 | 20.0 |
| | BSR | 68.8 | 54.1 | 74.3* | 67.2 | 28.3 | 40.8 | 46.3 | 46.4 | 53.3 | 15.8 |
| | CWT | **70.8** | **59.8** | **80.4*** | **71.5** | **34.3** | **47.0** | **53.7** | **52.6** | **58.8** | **15.0** |
| **DN-121** | DIM | 68.4 | 41.2 | 45.3 | 97* | 20.2 | 26.8 | 33.9 | 35.7 | 46.1 | 25.1 |
| | SIM | 74.8 | 45.5 | 49.6 | 98.6* | 20.9 | 30.1 | 38.4 | 40.6 | 49.8 | 25.2 |
| | Admix | 81.3 | 55.9 | 57.3 | **99.3*** | 28 | 35 | 43.9 | 48.1 | 56.1 | 23.8 |
| | MaskBlock | 67.5 | 35.3 | 39.1 | 95.1* | 15 | 22.7 | 29.3 | 31.7 | 42.0 | 26.4 |
| | US-MM | 82.0 | 54.4 | 58.6 | 98.7* | 27.5 | 35.2 | 45.8 | 47.8 | 56.4 | 23.8 |
| | BSR | **83.8** | 56.3 | 61.0 | 96.6* | 29.9 | 38.6 | 48.5 | 48.7 | 57.9 | 22.4 |
| | CWT | 83.1 | **60.9** | **64.7** | 98.6* | **35.1** | **46.2** | **52.9** | **54.8** | **62.0** | **20.4** |

Table A4: Attack success rates (%) of adversarial examples generated using various attack methods across eight classification models under **NRP**. The surrogate models are **Transformer-based**. * indicates the surrogate model.

| Model | Attack | RN-18 (↑) | RN-101 (↑) | RX-50 (↑) | DN-121 (↑) | ViT-B (↑) | PiT-B (↑) | ViF-S (↑) | Swin-T (↑) | Mean (↑) | Std. Dev. (↓) |
|---|---|---|---|---|---|---|---|---|---|---|---|
| **ViT-B** | DIM | 52.5 | 34.3 | 34.1 | 47.9 | 70.9* | 34.4 | 35.4 | 39 | 43.6 | 13.1 |
| | SIM | 57.6 | 36.1 | 39.2 | 52.5 | 85.3* | 38.4 | 40.5 | 47 | 49.6 | 16.3 |
| | Admix | 57.9 | 37.9 | 41.4 | 54.2 | **86.9*** | 40.7 | 41.8 | 51.3 | 51.5 | 16.0 |
| | MaskBlock | 56.8 | 30.3 | 32.7 | 47.2 | 81.6* | 30.1 | 32.6 | 40.6 | 44.0 | 17.9 |
| | US-MM | 61.1 | 40.0 | 42.8 | 58.6 | 84.6* | 42.5 | 44.7 | 55.0 | 53.7 | 15.5 |
| | BSR | 66.0 | 49.9 | 51.9 | 62.8 | 69.1* | 55.3 | 55.1 | 56.8 | 58.4 | **6.9** |
| | CWT | **66.7** | **52.1** | **53.8** | **64.1** | 75.9* | **59.1** | **57.7** | **58.8** | **61.0** | 7.7 |
| **PiT-B** | DIM | 56 | 32.2 | 36.1 | 48.1 | 27.2 | 67.8* | 37.2 | 38.1 | 42.8 | 13.5 |
| | SIM | 53.1 | 32 | 34.5 | 45.8 | 24.7 | 73.5* | 35.9 | 38.1 | 42.2 | 15.3 |
| | Admix | 54.8 | 34.7 | 36 | 47 | 25.8 | 73.8* | 39.8 | 42.8 | 44.3 | 14.7 |
| | MaskBlock | 56.8 | 29.7 | 34.2 | 44.5 | 23.1 | 76.5* | 34.8 | 37.1 | 42.1 | 17.2 |
| | US-MM | 68.4 | 37.4 | 39.2 | 52.2 | 29.4 | 74.4* | 43.6 | 46.0 | 47.6 | 14.0 |
| | BSR | 68.1 | 50.2 | 54.9 | 65.1 | 44.1 | 79.7* | 60.0 | 61.9 | 60.5 | 11.1 |
| | CWT | **70.6** | **56.7** | **59.9** | **69.3** | **52.7** | **85.1*** | **65.8** | **67.3** | **65.9** | **10.0** |
| **ViF-S** | DIM | 59.3 | 37.8 | 41.5 | 53.6 | 28.6 | 41.1 | 76.7* | 44.9 | 47.9 | 14.9 |
| | SIM | 60.7 | 38.2 | 40.3 | 53 | 27.6 | 41.2 | 80.8* | 47.8 | 48.7 | 16.4 |
| | Admix | 61.8 | 42.5 | 46.3 | 59.3 | 32 | 47.2 | 85.6* | 54.1 | 53.6 | 16.1 |
| | MaskBlock | 57.9 | 32.8 | 33.8 | 47.6 | 20.1 | 30.8 | 76.3* | 38.5 | 42.2 | 17.8 |
| | US-MM | 68.8 | 46.5 | 52.7 | 64.1 | 35.3 | 50.9 | 86.4* | 58.7 | 57.9 | 15.5 |
| | BSR | 71.7 | 52.1 | 56.8 | 69.2 | 40.5 | 59.3 | 81.6* | 61.4 | 61.6 | 12.7 |
| | CWT | **73.4** | **59.0** | **63.0** | **73.1** | **50.8** | **65.9** | **87.9*** | **68.9** | **67.8** | **11.1** |
| **Swin-T** | DIM | 54.6 | 31.5 | 33.2 | 46.1 | 24.6 | 33.5 | 37.4 | 74.5* | 41.9 | 16.1 |
| | SIM | 45.7 | 25.4 | 25.9 | 37.4 | 14 | 22 | 24.5 | 70.4* | 33.2 | 17.9 |
| | Admix | 50.9 | 25 | 26.7 | 40.5 | 15.2 | 23.8 | 27.7 | 75.2* | 35.6 | 19.4 |
| | MaskBlock | 50.1 | 24.1 | 24.9 | 38.3 | 13.5 | 20.8 | 24.9 | 69.1* | 33.2 | 18.4 |
| | US-MM | 52.1 | 26.1 | 29.6 | 43.1 | 16.2 | 23.2 | 28.3 | 74.3* | 36.6 | 19.0 |
| | BSR | 68.8 | 48.6 | 51.5 | 64.6 | 36.7 | 54.7 | 57.0 | 76.9* | 57.4 | 12.6 |
| | CWT | **70.9** | **54.6** | **57.7** | **71.5** | **45.1** | **62.4** | **62.6** | **83.1*** | **63.5** | **11.7** |

regions of the object. This two-step process effectively balances attention distribution and enhances the model's robustness.

Moreover, the choice of interpolation method plays a critical role in shaping attention distributions. **Bicubic interpolation**, while smooth, often over-smooths attention maps, making it less effective at capturing distinct object regions. **Nearest neighbor interpolation** demonstrates insensitivity to scaling factors above 1.0. **Area interpolation**, on the other hand, is overly sensitive to scaling factors below 1.0, resulting in attention maps that collapse onto irrelevant regions and fail to preserve essential object features. In contrast, **bilinear interpolation** achieves a balance between smoothness and precision, producing the most consistent and well-distributed attention maps.

Thus, we adopt **bilinear interpolation** in our method to ensure optimal attention distribution and robust adversarial performance.

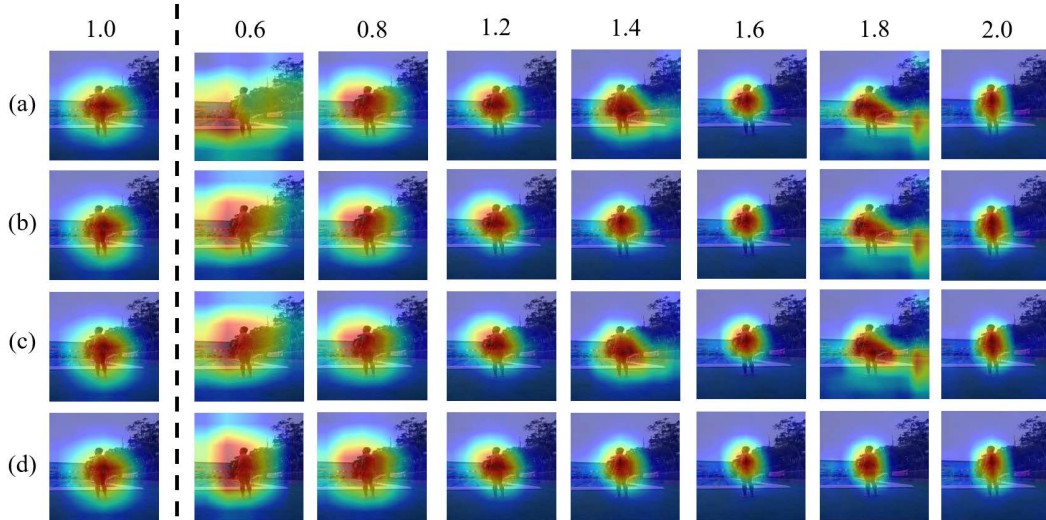

Figure A1: **Heatmaps of Different Interpolation Rates Using ResNet-18**. (a) Area interpolation; (b) Bicubic interpolation; (c) Bilinear interpolation; (d) Nearest-neighbor interpolation.

## A.6 VISUALIZATION FOR SEMANTIC CONSISTENCY

Fig. A2 showcases adversarial examples generated by CWT and BSR. As the figure illustrates, while achieving a high ASR, CWT effectively preserves the semantic information and visual distinguishability of the original image.

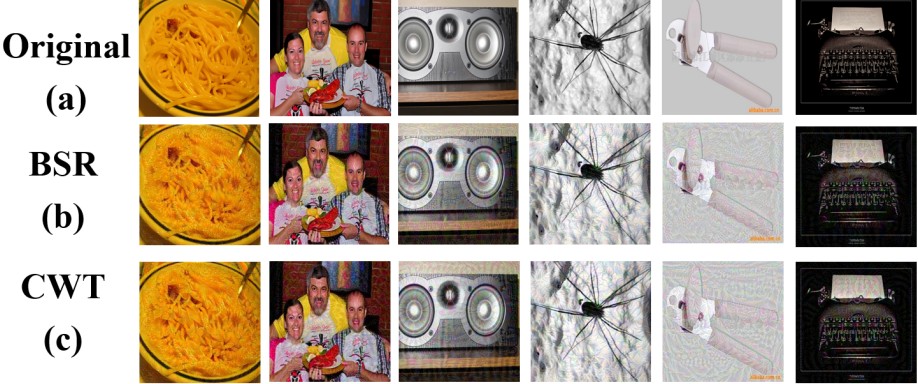

Figure A2: **Viusalization for Semantic Consistency**.

## A.7 DEFENSE METHOD SETUPS

**AT:** We employ the pre-trained ResNet-50 model from the official `fast-4px` configuration.

**DiffPure:** We utilize the official implementation with a ResNet-101 classifier. The key hyperparameters are set as follows: diffusion timestep $t = 150$ and perturbation budget $\epsilon = 4/255$.

**HGD:** We construct the defense by loading all four pre-trained denoising models (`denoise_res`, `denoise_inres`, `denoise_incepv3`, and `denoise_rex`) provided in the official repository.

**RS:** This method uses a ResNet-50 as the base classifier, with a Gaussian noise standard deviation of $\sigma = 0.50$.

## A.8 COMPARISON OF CWT WITH DIM AND BSR

DIM enhances the transferability of adversarial attacks by applying resizing and padding to input samples, whereas BSR achieves this by randomly shuffling and rotating image blocks. Despite incorporating similar operations, our method, CWT, differs significantly from these approaches. Below, we highlight the distinctions between our method and the two methods.

- **CWT vs. DIM** In contrast to DIM, which performs scaling with fixed values on the entire image, our method emphasizes a random enlargement operation with pre-interpolation. Moreover, while DIM applies this operation to the entire image uniformly, our approach targets individual blocks within the image, introducing more localized transformations.

- **CWT vs. BSR** While both methods employ block partitioning, CWT makes several critical departures from BSR. First, CWT replaces BSR's shuffle operation with block-wise interpolation to alter the image structure. The shuffle operation can drastically disrupt the semantic structure of an object by altering the relative positions of its parts (e.g., placing a cat's paw on its head) and transposing foreground features onto what were originally background regions. Such severe alterations can lead to the computation of ineffective or even counterproductive gradients. In contrast, CWT's localized interpolation preserves the relative positions of object components while modifying their relative sizes, maintaining greater semantic consistency. Second, CWT adopts a selective rotation mechanism, applying it to only a limited subset of blocks. While rotation can cause information loss, CWT mitigates this effect by performing the rotation on the enlarged blocks, thus minimizing the impact on critical feature details.

## A.9 EXTENDED EVALUATIONS ON VLMS

To further assess the robustness of CWT adversarial examples, we tested them on state-of-the-art Vision-Language Models (VLMs), including GPT-5 [2] and Gemini Pro [3]. As illustrated in Fig. A3, the adversarial perturbations lead to significant failures in both object recognition and semantic description. For instance, the models fail to identify a common wall socket, instead describing it merely as a "small rectangular patch" or "yellowish patterned fabric". More severe errors include misinterpreting a cat as a "surreal, dreamlike image" where the subject is "unrecognizable", or grotesquely misclassifying it as a "dead or dying canid" in a "graphic and disturbing" scene.

---

[2] https://chatgpt.com/
[3] https://gemini.google.com/

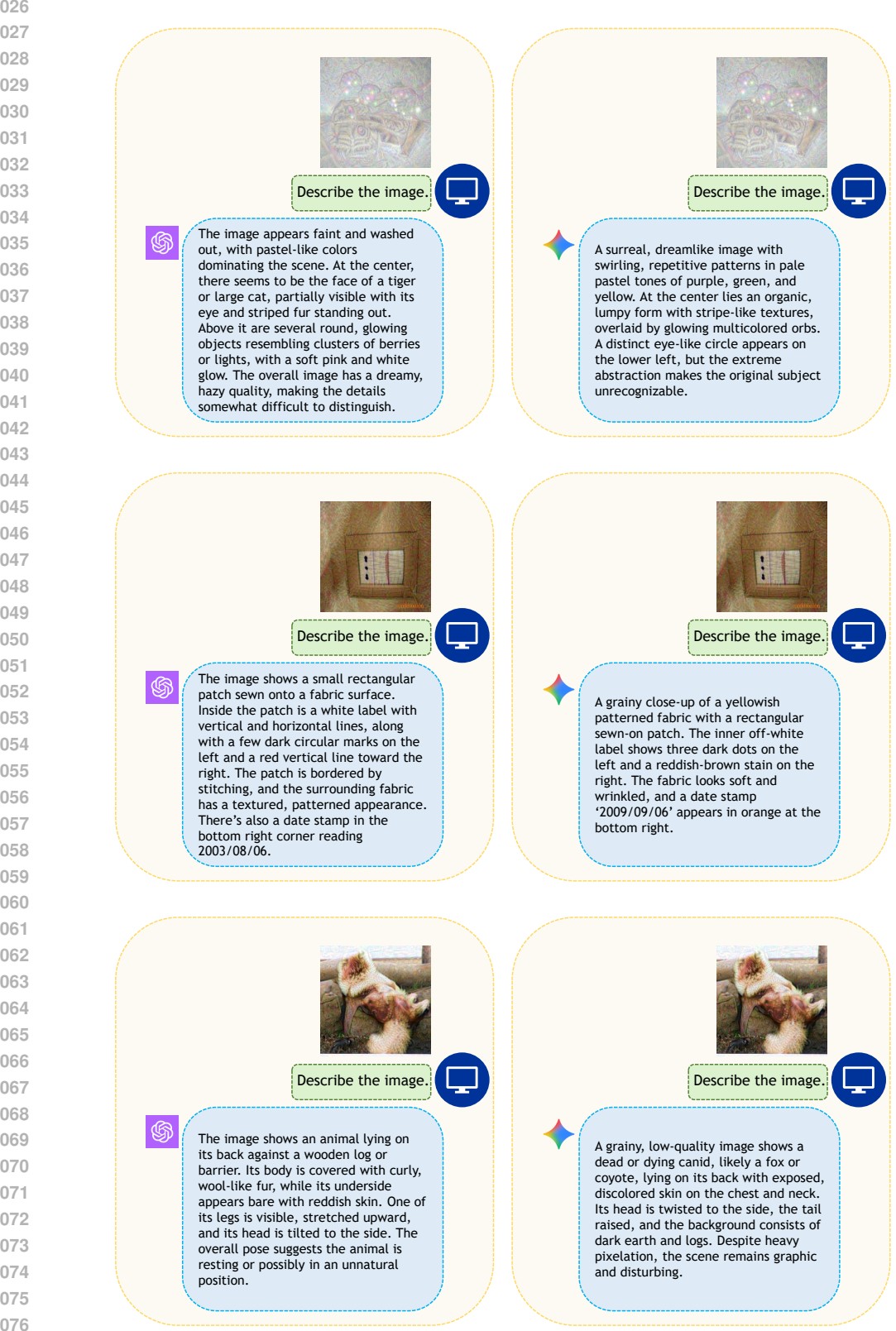

Figure A3: Extended Evaluations on GPT-5 and Gemini Pro.

