# OpenReview forum: "Enhancing Adversarial Transferability via Component-Wise Transformation"
_ICLR.cc/2026/Conference — ICLR 2026 Conference Withdrawn Submission_

### Official Review · Reviewer_PLGw · 2025-10-29

**Soundness:** 3
**Presentation:** 3
**Contribution:** 2
**Rating:** 4
**Confidence:** 4

**Summary:**

The paper proposes Component‑Wise Transformation (CWT), a block-wise input transformation method to improve adversarial example transferability across architectures.

**Strengths:**

1 Cross-architecture adversarial transfer is a relevant and challenging research topic. Grad-CAM visualizations convincingly illustrate architectural attention differences.

2 CWT is easy to implement, computationally efficient, and integrates naturally with MI-FGSM.

**Weaknesses:**

1 Incomplete comparison with SOTA: The paper omits some SOTA methods (e.g., Boosting the Transferability of Adversarial Attack on Vision Transformer with Adaptive Token Tuning, Improving Adversarial Transferability via
Intermediate-level Perturbation Decay),

2 There is no rigorous quantitative analysis showing that CWT systematically improve gradient alignment, or that improved alignment directly leads to higher transfer success. Without this connection, it is unclear whether the empirical gains are due to the claimed mechanism or just incidental effects of more diverse transformations.

**Questions:**

1 Have you evaluated targeted transfer attacks or adaptive defenses?

2 Can you provide more empirical evidence that CWT consistently increases gradient alignment across architectures?

---

### Official Review · Reviewer_cTxy · 2025-10-30

**Soundness:** 2
**Presentation:** 3
**Contribution:** 1
**Rating:** 2
**Confidence:** 5

**Summary:**

This paper works on improving the input transformations directions in image adversarial attacks, The paper first explores the theoretical roots of poor cross-architecture transferability, and then proposes component-wise transformation (CWT) aiming to boost the transferability. The authors conducted. extensive experiments on standard datasets like ImageNet and COCO.

**Strengths:**

- The paper is easy to follow.
- The paper studies an important field in AI security.

**Weaknesses:**

1. Evaluation.
    - Model utility is not included in evaluations. The reviewer is concerned that block-wise rotation may include artifacts in image, such that the artifacts could be easily observed by a human, hence it may hinder the **imperceptibility**, a pivotal principle in adversarial attack field. The reviewer kindly asks the authors to provide more examples of adversarial examples generated by the proposed algorithm to showcase their imperceptibility**.**
2. Soundness of the method
- It is not clear that the top part of figure 2 is based on real data, or hand-made illustrations.
- On Equation (6): the claims in lines 209-211 could be essentially wrong. The reason is that as shown in equation (6), two gradient operators are regarding different inputs, i.e., $x_s^{adv}$ and $x$ (another concern: $x$ is not defined here).
- The IoU metric proposed in Equation (7) may be problematic. The equation builds upon the assumption that every images contains a foreground (and hence a background), however it is not clear that the assumption holds in the scenario in the paper.

3. Presentation

- For Figure 2, the meaning of “attack directions from” is not clear.
- The term 'attention' in Figure 2 is misleading. The reviewer recommends replacing it with 'attribution map' or 'saliency map' (for CAM methods) to prevent confusion with the attention operations used in Transformer.
- Figure 3. should appear in the evaluation section (i.e., section 4) instead of methodology section.
- $s$ and $t$ should be italicized in lines 193-194.

**Questions:**

Please respond the concerns above.

The reviewer will raise the rating if the concern gets solved.

---

### Official Review · Reviewer_XAUX · 2025-11-02

**Soundness:** 2
**Presentation:** 2
**Contribution:** 2
**Rating:** 2
**Confidence:** 4

**Summary:**

This paper proposes Component-Wise Transformation (CWT) to boost adversarial transferability. CWT applies interpolation and selective rotation to individual image blocks, ensuring that each transformed image highlights different target regions. Extensive experiments on the ImageNet and COCO datasets demonstrate that CWT consistently outperforms state-of-the-art methods.

**Strengths:**

The experiment scope on the COCO dataset is good.

The topic of adversarial transferability is important.

**Weaknesses:**

The motivation is not convincing.

The details of the proposed method are unclear.

The experiment may be unfair.

(Minor) Format issue in Table 2.

**Questions:**

1 The authors point out that the previous approaches fail to alter the model’s attention, leading to an attention distribution remaining consistent with the original image or deviating from the areas of interest in Lines 101-104. However, why this attention can influence the adversarial transferability is not discussed.

2 CWT encourages models to focus on diverse regions of the object in the original image, which can enhance the transferability in Lines 105 - 107. However, why focusing on a diverse part of the object can improve the transferability is not validated.

3 In Lines 139-142, the authors aim to improve the attention diversity by the proposed CWT. However, in the motivating example Table 1, the authors measure the IoU as the metric to compare with BSR. The IoU only measures the coverage, not the diversity. Therefore, the motivating example is not convincing enough to demonstrate the effect of diversity on performance.

4 In the method section, the content of the Component-wise Transformation approach is less than one page. The details of the approach are missing. The authors should show the workflow of their approach to visualize the intermediate transformed images of their approach. Otherwise, the semantic meaning of the transformed image cannot be guaranteed.

5 In Algorithm 1, I cannot find any useful information. The core part of the CWT approach is covered by only one sentence, “Calculate the gradient by Eq. 11.” The remaining part is the general optimization framework, similar to other attacking algorithms.

6 The details of other baselines should be discussed in the experimental setup to make sure you can compare with them in a fair way. E.g., you can state the scaled copies of other baselines for a fair comparison under a similar computation complexity. Otherwise, your Runtime experiment is also unfair.

---

### Official Review · Reviewer_fMuc · 2025-11-02

**Soundness:** 3
**Presentation:** 3
**Contribution:** 2
**Rating:** 4
**Confidence:** 4

**Summary:**

This paper focuses on enhancing the transferability of adversarial examples under black-box models. The authors propose a method called CWT (Component-Wise Transformation), which applies interpolation and selective rotation to individual image blocks to guide the model's attention toward different target regions. The paper is well written and easy to read, and the experimental results have validated the effectiveness of the method.

**Strengths:**

1. This paper reveals that the differences in the target regions focused on by various models may be a primary reason for the limited transferability of adversarial examples.

2 . Fig.3 shows the relationship between different models from three perspectives: gradient cosine similarity, adversarial noise cosine similarity, and transferability of attacks.

3. The experimental results of the CWT method demonstrate the effectiveness of the method proposed in this paper.

4. Fig.A3 in the appendix demonstrates the effectiveness of the CWT method against attacks in real-world scenarios.

**Weaknesses:**

1. In Figure 2, the authors claim that the BSR method focuses too much on the region outside the target object, i.e., the background area, which limits the transferability of adversarial examples. Does this description contradict the actual experimental results? Methods like DIM, SIM, and Admix, even though they focus on the region within the target object, exhibit worse transferability than BSR. Furthermore, since the model's input is a whole image rather than just the target object, attackers can enhance perturbations in background regions outside the target to enhance the attack capability of the entire photo. Therefore, I think this part of the description is overly absolute.

2. The CWT method seems to be an integration of multiple input transformation techniques (Pre-Interpolation, Block-wise Scaling, Rotation). The authors should theoretically elaborate on the positive effects each input transformation method brings to guiding the model's focus on different regions of the target object.

3. I think the comparison in Table 1 is not a fair comparison. Figure 2 already shows that the region of interest for BSR is not on the target object. The authors should calculate the IOU results with methods like DIM, SIM, and Admix to highlight your motivation, i.e., “Aims to generate transformed images that strategically encourage models to focus on diverse regions of an object.”. Additionally, from Figure 2, I don't clearly see any differences in the regions of interest between the CWT method and those used by DIM, SIM, and Admix.

4. For the third contribution, since the authors only compared some input transformation-based methods, it is recommended to revise it to “compared to existing input transformation-based approaches.”

5. Line 266 seems to have an incorrect word.

**Questions:**

Please refer to the weaknesses section.

---

### Note · Authors · 2025-11-20

**Comment:**

I have read and agree with the venue's withdrawal policy on behalf of myself and my co-authors.

**Withdrawal Confirmation:**

I have read and agree with the venue's withdrawal policy on behalf of myself and my co-authors.